# Nativity, Race, Ethnicity, and Hypertensive Disorders During Pregnancy

**DOI:** 10.3390/jcm14134594

**Published:** 2025-06-28

**Authors:** Madeline R. Fram, Jeff M. Denney, Kristen H. Quinn

**Affiliations:** Section on Maternal-Fetal Medicine, Department of Obstetrics and Gynecology, Wake Forest School of Medicine, Winston-Salem, NC 27157, USA; mfram@uthsc.edu (M.R.F.); krquinn@wakehealth.edu (K.H.Q.)

**Keywords:** pregnancy, nativity, race, ethnicity, hypertension in pregnancy, pregnancy

## Abstract

**Background:** Previous research has examined the idea of the healthy immigrant effect and its potential application to pregnancy-related hypertensive disorders, particularly inracial and ethnic minority groups. The aim of this study is to examine how nativity, race, and ethnicity are related to prevalence of pregnancy-related hypertensive disorders. **Methods:** A retrospective cohort study was conducted using data from the PRAMS CDC dataset. These data were analyzed via *Χ*^2^ comparisons of prevalence of pregnancy-related hypertensive disorders in foreign and US-born mothers, including subgroup analyses for Black and Hispanic women. **Results**: A sample size of 63,648 was analyzed, and revealed significant differences in the prevalence of gestational hypertension between US-born vs. foreign-born Black mothers (12.6% vs. 8.0%, *Χ*^2^ (1, N = 12,046) = 36.92, *p* < 0.001), Hispanic mothers (9.5% vs. 7.2%, *Χ*^2^ (1, N = 11,524) = 18.236, *p* < 0.001), and the larger sample of mothers across all reported racial and ethnic backgrounds (11.0% vs. 7.0%, *Χ*^2^ (1, N = 63,648) = 163.835, *p* < 0.001). The results also revealed a significant difference in the prevalence of hypertension eclampsia between US-born and foreign-born Hispanic mothers (0.8% vs. 0.3%, *Χ*^2^ (1, N = 11,152) = 8.480, *p* = 0.004). **Conclusions:** The study results showed evidence of significantly lower prevalence of pregnancy-related hypertensive disorders among foreign-born mothers as compared to their US-born counterparts, both in the full study sample and specifically in the subsamples of women who self-identified as Black and women who self-identified as Hispanic. These variances by nativity, race, and ethnicity provide further insight into how the healthy immigrant effect can apply to pregnancy-related hypertensive disorders, particularly for women of racial and ethnic minorities.

## 1. Introduction

Maternal mortality and high-risk pregnancy are topics that deserve to be the focus of continued research and efforts to decrease their incidence and reduce the associated morbidity and mortality. Pregnancy-related hypertensive disorders have been cited as the cause of 7.8% of pregnancy-related mortalities in the United States [1,2]. Particularly striking is the disproportionate impact of pregnancy-related hypertensive disorders on patients of racial and ethnic minorities [3,4]. 

Preeclampsia deserves significant attention due to its place as a leading worldwide cause of maternal mortality and morbidity [5]. In a morbidity and mortality report analyzing data from 2007 to 2016, 7.8% of pregnancy-related mortalities in the United States were attributed to pregnancy-related hypertensive disorders [1,2]. One of the aspects of maternal mortality in the United States, and specifically preeclampsia, that continues to be studied is its disproportionate impact on patients of racial and ethnic minorities [3]. The aforementioned morbidity and mortality report also showed a significantly higher proportion of pregnancy-related deaths attributed to pregnancy-related hypertensive disorders among Black women than among White women [1]. These racial disparities exist to such an extent that “Black race” is listed as a moderate risk factor for developing preeclampsia on the ACOG (American College of Obstetricians and Gynecologists) website [6,7]. Alongside this racial disparity, there is a conversation about the difference in rates of preeclampsia between Hispanic women and White women. Unlike the previous research regarding higher rates of preeclampsia among Black women, this research is a bit more variable [8]. 

While some studies show higher rates of preeclampsia among Hispanic women as compared to their White counterparts, others do not [8]. A study using data from New York City between the years 1995–2003 broke down their Hispanic population into subgroups by region and found that Mexican women had the highest risk of preeclampsia among major ethnic groups in their study [8,9]. This is an example of a study which provides further insight into the prevalence of preeclampsia among Hispanic women in the United States by examining Hispanic women by country rather than as a demographic monolith.

This study aims to conduct a deeper dive into rates of preeclampsia for women of racial and ethnic minorities who are living in the United States, with a particular focus on women who were born outside of the United States but have moved to the US. Existing literature suggests the idea of the “healthy immigrant effect”—the phenomenon of immigrants new to the United States initially demonstrating better health metrics than their US-born counterparts, but declining as their tenure in the United States increases [10]. Previous literature indicates that the amount of time that immigrants who identify as Black and Hispanic spend in the United States has a greater negative impact on their health as compared to White and Asian immigrants [10]. This observed phenomenon leads to questions both about what aspects of health this effect can be observed in and about why this association exists.

One area of health in which the healthy immigrant effect has been studied is pregnancy-related hypertensive disorders. A study using the 2014–2015 national birth cohort data found evidence of lower rates of maternal hypertension amongst immigrant women as compared to US-born women [11]. A 2021 study conducted by Boakye et al. examines the prevalence of preeclampsia in foreign-born non-Hispanic Black women as compared to US-born non-Hispanic Black women, and found lower rates in the population of foreign-born non-Hispanic black women [12]. Furthermore, this study found that among foreign-born non-Hispanic Black women who had lived in the US for ≥10 years, there was no longer a significant decrease in their odds of having preeclampsia when compared to US-born non-Hispanic Black women [12]. These studies bring to light an important aspect of the conversation about the disproportionate rates of maternal mortality for patients of racial and ethnic minorities. Future and current research is important to identify unintentional effects of policies or practices that may restrict immigrant communities’ access to social services. This important piece of the conversation centers around the variable of nativity.

The present study aims to use this variable of nativity to further examine the relationship between high-risk pregnancies and women in racial and ethnic minority groups. Specifically, this study will look at pregnancy-related hypertensive disorders. A CDC dataset from the Pregnancy Assessment Monitoring System was used to draw the following comparisons with respect to proportion of gestational hypertension and eclampsia (1) amongst women in the United States who are born outside of the US to US-born women; and, (2) amongst women who have self-identified as Black and as Hispanic. The aim of this study is to add further information to the conversation about racial and ethnic inequities and the healthy immigrant effect and how these aforementioned issues impact the health of pregnant women.

## 2. Materials and Methods

This is a retrospective cohort study utilizing data from the Pregnancy Assessment Monitoring System (PRAMS) Automated Research File (ARF) Portal. Wake Forest University School of Medicine Institutional Review Board (Medical Center Blvd, Winston Salem, North Carolina) approved our study “**High Risk Pregnancy Rates for Non U.S. Born Mothers**” IRB00110746 (2 February 2023). Our investigation was conducted in accordance with the principles outlined in the Declaration of Helsinki to ensure adherence to national and international guidelines.

Upon approval by our IRB, data was generated from the Phase 8 study; said data was collected between 2016 and 2021. The PRAMS data came from a Centers for Disease Control (CDC 1600 Clifton Rd., Atlanta, Georgia) and site health department collaboration in which women who had recently undergone childbirth were selected via the infants’ birth certificate(s) and then contacted by mail to generate survey data. PRAMS was, in brief, a mixed-mode data collection method in which 1–3 questionnaires were mailed to respondents to collect information on the maternal health, behaviors, and experiences of women who have given birth in the US. Non-responders are followed by attempted telephone interviews.

Multiple health department sites participated across the United States in effort to collect data, which was relevant to the study aims [4]. Responses were self-reported and generated solely by women who had experienced a live birth. Subjects were stratified by their response (‘yes’ or ‘no’) with regard to whether they had experienced gestational hypertension or hypertension eclampsia. Note that this paper utilizes the nonstandard term “hypertension eclampsia” because this is the term utilized in the PRAMS dataset from which this study’s conclusions were drawn. Accordingly, eclampsia is herein referred to as hypertension eclampsia based on PRAMS item labels.

For this study, SPSS (Version 27.0, Armonk, NY, USA) was used to analyze the proportions of gestational hypertension and hypertension eclampsia among mothers who were born outside of the United States and mothers who were US-born [13]. These comparisons were made using chi square tests. A total of 6 chi square (*Χ*^2^) tests were run using this dataset. The first was a chi square comparing proportions of gestational hypertension among mothers born in the US to proportions of gestational hypertension among mothers born outside of the US. The second was a chi square comparing rates of hypertension eclampsia among mothers who were born in the US to proportions of hypertension eclampsia among mothers who were born outside of the US. In order to compare the proportions of gestational hypertension and hypertension eclampsia amongst Black mothers who were born outside of the US and Black mothers who were US-born, the aforementioned tests were repeated on a subset of the data including only mothers who self-identified their race as Black. In order to compare proportions of gestational hypertension amongst Hispanic mothers who were born outside of the US and Hispanic mothers who were US-born, the aforementioned tests were then repeated on a subset of the data including only mothers who self-identified their ethnicity as Hispanic.

## 3. Results

Of the participants who responded either yes or no when asked whether they had ever been diagnosed with gestational hypertension, 51,993 reported having been born in the United States and 11,655 reported having been born outside of the United States, yielding a total sample of 63,648 participants. Of these participants, 6525 reported receiving a diagnosis of gestational hypertension and 57,123 reported not receiving a diagnosis of gestational hypertension. Meanwhile, 11.0% [n = 5709] of this population of mothers born in the US had a diagnosis of gestational hypertension, which was significantly higher than the 7.0% [n = 816] of this population of mothers not born in the US who had a diagnosis of gestational hypertension, *Χ*^2^ (OR 6.28 CI 5.84–6.76 for mothers not born in US with gestational HTN vs. mothers born in the US *p* < 0.001); see Table 1.

Of the participants who responded either yes or no when asked if they had ever been diagnosed with hypertension eclampsia, 51,354 reported having been born in the United States and 10,897 reported having been born outside of the United States, yielding a total sample of 62,251 participants. Of these participants, 460 reported being diagnosed with hypertension eclampsia and 61,791 reported that they had not been diagnosed with hypertension eclampsia. Meanwhile, 0.7% [n = 369] of this population of mothers born in the US had a diagnosis of hypertension eclampsia, which was not significantly lower than the 0.8% [n = 91] of this population of mothers not born in the US who had a diagnosis of hypertension eclampsia, *Χ*^2^ (1, N = 62,251) (OR 1.16 CI 0.92–1.46 for mothers not born in US vs. US-born mothers who had hypertension eclampsia; *p* = 0.197); see Table 2.

Of the participants who self-identified their race as Black and who responded either yes or no when asked if they had ever been diagnosed with gestational hypertension, 9781 reported having been born in the United States and 2265 reported having been born outside of the United States, yielding a total sample of 12,046 participants. Of these participants, 1414 reported having been given a diagnosis of gestational hypertension and 10,632 reported not having been diagnosed with gestational hypertension. Meanwhile, 12.6% [n = 1232] of this population of self-identified Black mothers born in the US had a diagnosis of gestational hypertension, which was significantly higher than the 8.0% [n = 182] of this population of self-identified Black mothers not born in the US who had a diagnosis of gestational hypertension, *Χ*^2^ (1, N = 12,046); an OR of 0.69 and CI 0.59–0.82 were reported for Black mothers not born in the US who were not diagnosed with GHTN vs. Black mothers born in the US who were diagnosed with GHTN, *p* < 0.001; see Figure 1.

Of the participants who self-identified their race as Black and who responded either yes or no when asked if they had ever been diagnosed with hypertension eclampsia, 9627 reported having been born in the United States and 1966 reported having been born outside of the United States, yielding a total sample of 11,593 participants. Of these participants, 77 reported having been diagnosed with hypertension eclampsia and 11,516 reported not having been diagnosed with hypertension eclampsia. Meanwhile, 0.6% [n = 59] of this population of self-identified Black mothers born in the US had previously been diagnosed with hypertension eclampsia, which was not significantly lower than the 0.9% [n = 18] of this population of self-identified Black mothers not born in the US who had previously been diagnosed with hypertension eclampsia, *Χ*^2^ (1, N = 11,593); an OR of 1.49 and CI of 0.88–2.53 were reported for Black mothers who were not born in the US and had a history of hypertension eclampsia vs. Black mothers born in US who had a history of hypertension eclampsia, *p* = 0.132.

Of the participants who self-identified their ethnicity as Hispanic and who responded either yes or no when asked if they had ever been diagnosed with gestational hypertension, 6987 reported having been born in the United States and 4537 reported having been born outside of the United States, yielding a total sample of 11,524 participants. Of these participants, 990 reported having previously been diagnosed with gestational hypertension and 10,534 reported that they had never been diagnosed with gestational hypertension. Meanwhile, 9.5% [n = 663] of this population of self-identified Hispanic mothers born in the US had previously been diagnosed with gestational hypertension, which was significantly higher than the 7.2% [n = 327] of this population of self-identified Hispanic mothers not born in the US who had a diagnosis of gestational hypertension, *Χ*^2^ (1, N = 11,524); an OR of 1.31 and CI of 1.14–1.51 were reported for Hispanic mothers born in US vs. Hispanic mothers not born in US with regard to their likelihood of experiencing gestational hypertension *p* < 0.001; see Figure 2.

Of the participants who self-identified their ethnicity as Hispanic and who responded either yes or no when asked if they had ever been diagnosed with hypertension eclampsia, 6846 reported having been born in the United States and 4306 reported having been born outside of the United States, yielding a total sample of 11,152 participants. Meanwhile, 66 of these participants also reported having been diagnosed with hypertension eclampsia and 11,086 reported that they had not been diagnosed with hypertension eclampsia. 

Additionally, 0.8% [n = 52] of this population of self-identified Hispanic mothers born in the US had been diagnosed with hypertension eclampsia, which was significantly higher than the 0.3% [n = 14] of this population of self-identified Hispanic mothers not born in the US who had a diagnosis of hypertension eclampsia, *Χ*^2^ (1, N = 11,152); an OR of 2.34 CI 1.29–4.2 was observed for Hispanic mothers born in the US vs. Hispanic mothers not born in US in terms of their likelihood of being diagnosed with hypertension eclampsia, *p* = 0.004; see Figure 3.

## 4. Discussion

This research found further evidence of differences in the prevalence of pregnancy-related hypertensive disorders amongst foreign-born mothers as compared to their US-born counterparts. This phenomenon was found both in the two minority groups analyzed in this study (Black race and Hispanic ethnicity) and in the dataset including participants of all self-reported races and ethnicities. Although the aim of this study was primarily to examine the prevalence of pregnancy-related hypertensive disorders amongst mothers in racial and ethnic groups which have higher rates of these disorders in the US, it was interesting to see evidence of this healthy immigrant effect across participants from all racial and ethnic backgrounds. This is an important finding because it opens up the discussion about risk factors for developing pregnancy-related hypertensive disorders to include an examination of risk factors that are both specific to residents of the United States but are not specific to any particular demographic within the US population.

 Looking specifically at the first subsample examined in this study—participants who self-identified their race as Black—the finding appears to replicate findings in the previous study referenced earlier [12]. Like this study, Boakye et al.’s study found a lower prevalence of pregnancy-related hypotensive disorder (their study specifically looked at preeclampsia) in their foreign-born sample of Black women as compared to their US-born counterparts [12]. A piece of information that the present study adds to the conversation is this replicated effect using a national dataset, as Boakye et al. utilized a dataset from the Boston Birth Cohort. The present study’s finding of a lower prevalence of a hypertensive disorder of pregnancy amongst Black foreign-born women as compared to their US-born counterparts on a national level is reflective of the work Singh et al. presented in their study using national birth cohort data between 2014 and 2015 [11]. Overall, this study adds further evidence to suggest that the healthy immigrant effect can be appropriately applied to explain higher rates of pregnancy-related hypertensive disorders for US-born Black women than for their foreign-born counterparts.

 The second subsample examined in this study is made up of participants who self-identified their ethnicity as Hispanic. As mentioned before, there is less consensus about the relationship between Hispanic ethnicity and prevalence of pregnancy-related hypertensive disorders [8]. Some previous work has shown evidence of higher risks of preeclampsia for subgroups of Hispanic women [8,9]. However, other prior work has concluded that disparities in prevalence of preeclampsia could not be attributed to nativity for Hispanic women [14]. The present study adds to this discussion by providing evidence of significantly lower rates of both gestational hypertension and hypertension eclampsia for foreign-born Hispanic women than for their US-born counterparts.

The analysis focused on this group was the only one which yielded a significant difference in prevalence of hypertension eclampsia between the foreign-born participants and that of their US-born counterparts. The overall subset of participants with hypertension eclampsia (460) is much smaller than the subset of participants with gestational hypertension (6535). Consequently, drawing a strong conclusion is difficult when basing it on the significantly greater proportion of US-born Hispanic women with hypertension eclampsia as compared to their foreign-born counterparts in the context of the lack of a significant difference for this comparison in our groups of Black women and the full dataset of women. It does suggest a potentially uniquely lower risk of hypertension eclampsia for foreign-born Hispanic mothers, but further research should be conducted before this conclusion is made.

Although this study does provide further insight into the conversation about the relationships between nativity, race, ethnicity, and pregnancy-related hypertensive disorders, as it aimed to do, it also brings up some unanswered questions. These questions remain unanswered by this study due to the limitations of the secondary data used for the analyses. This dataset did not include information about two variables of particular interest—length of tenure in the United States and country of origin.

Due to this limitation, this study does not address how length of tenure in the United States impacts a person’s risk of developing pregnancy-related hypertensive disorders. Previous studies about the healthy immigrant effect suggest a diminished association with better health in foreign-born persons as the amount of time they have spent living in the US increases, with a greater decrease in the apparent protection or association observed with foreign nativity for Black and Hispanic immigrants [10]. However, the referenced study did not specifically examine the health of pregnant patients. Moreover, it would be an interesting follow-up to the present study to examine pregnancy-related hypertensive disorders in the context of length of tenure in the US for foreign-born mothers from racial and ethnic minority groups.

Another question left unanswered by this study pertains to a comparison of the prevalence of pregnancy-related hypertensive disorders amongst foreign-born US-residing mothers to the prevalence of pregnancy-related hypertensive disorders for women living in the countries where the foreign-born US-residing mothers are from. Because we do not have data regarding the countries of birth for our sample of mothers born outside of the United States, we were unable to compare the prevalence of pregnancy-related hypertensive disorders for our sample to the rates of their own countries. This would be an interesting future study because it would provide insight into whether the lower prevalence of pregnancy-related hypertensive disorders for the foreign-born population represented something unique about the population of mothers living in the US who are foreign-born or if it provided evidence of their retaining levels of prevalence similar to those of their native countries.

Overall, the present study echoes evidence of the healthy immigrant effect in pregnant mothers as seen in prior research and offers specific insight into this association as it relates to pregnancy-related hypertensive disorders. However, it is important to note that the relationships highlighted in this study cannot be used to infer causality due to the nature of the retrospective cohort methodology. In addition to contributing the direct results of our research, our study raises important questions that warrant consideration in future research efforts.

## Figures and Tables

**Figure 1 jcm-14-04594-f001:**
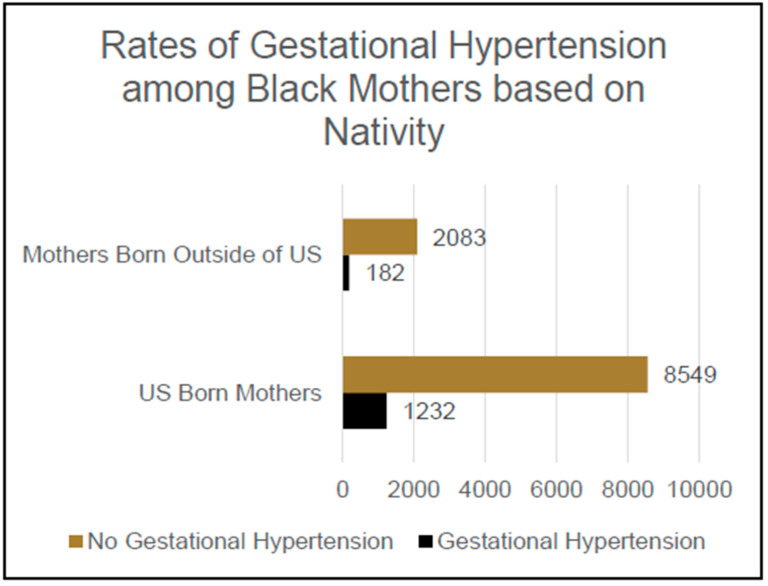
Rates of gestational hypertension among Black mothers based on nativity.

**Figure 2 jcm-14-04594-f002:**
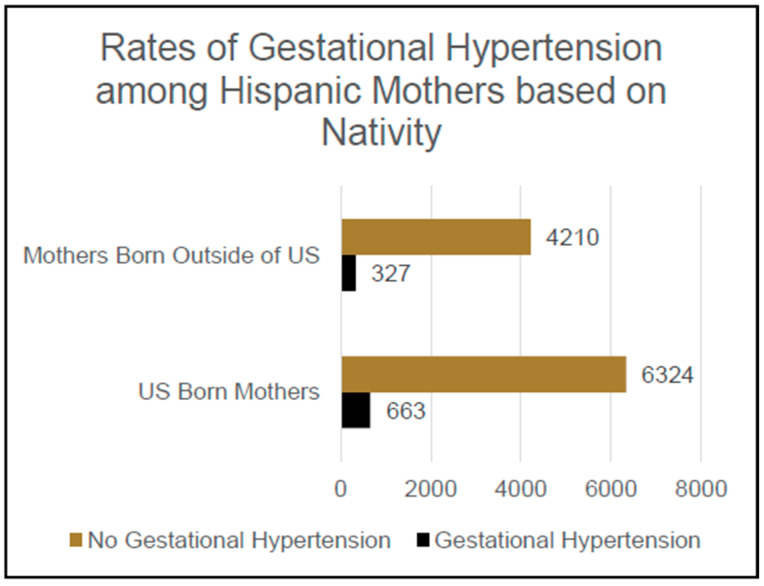
Rates of Gestational Hypertension among Hispanic Mothers based on Nativity.

**Figure 3 jcm-14-04594-f003:**
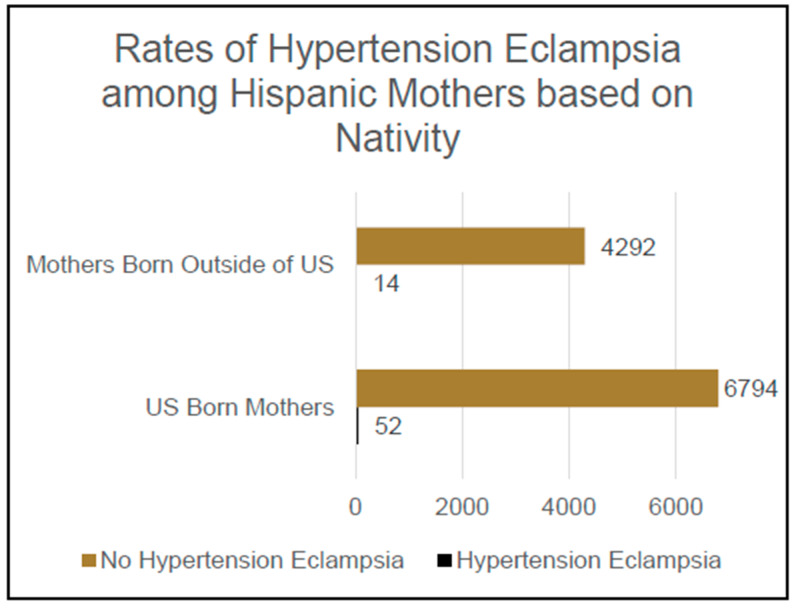
Rates of hypertension eclampsia among Hispanic mothers based on nativity.

**Table 1 jcm-14-04594-t001:** Gestational hypertension prevalence amongst foreign and US-born mothers.

	Gestational HTN	No Gestational HTN
**Full Dataset:**		
US-born	5709 (11.0%) *	46,284 (89.0%)
Foreign-born	816 (7.0%) *	10,839 (93.0%)
**Black Race:**		
US-born	1232 (12.6%) *	8549 (87.4%)
Foreign-born	182 (8.0%) *	2083 (92.0%)
**Hispanic Ethnicity:**		
US-born	663 (9.5%) *	6324 (90.5%)
Foreign born	327 (7.2%) *	4210 (92.8%)

* denotes significant difference.

**Table 2 jcm-14-04594-t002:** Hypertension eclampsia prevalence amongst foreign and US-born mothers.

	HTN Eclampsia	No HTN Eclampsia
**Full Dataset:**		
US-born	369 (0.7%)	50,985 (99.3%)
Foreign-born	91 (0.8%)	10,806 (99.2%)
**Black Race:**		
US-born	59 (0.6%)	9568 (99.4/%)
Foreign-born	18 (0.9%)	1948 (99.1%)
**Hispanic Ethnicity:**		
US-born	52 (0.8%) *	6794 (99.2%)
Foreign born	14 (0.3%) *	4292 (99.7%)

* denotes significant difference.

## Data Availability

Data may be made available upon request.

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
