# Peer review of "Nativity, Race, Ethnicity, and Hypertensive Disorders During Pregnancy"

_jcm, 2025, doi:10.3390/jcm14134594_

Round 1
Reviewer 1 Report
Comments and Suggestions for Authors
Reviewer Comments to the Authors
General Assessment
This manuscript addresses an important topic in maternal health, specifically the prevalence of hypertensive disorders of pregnancy (HDP) with respect to nativity, race, and ethnicity. Using the PRAMS dataset allows for a robust and meaningful analysis that supports the concept of the 'healthy immigrant effect.' The manuscript is generally well written, and the results are clearly presented. However, some areas would benefit from clarification, additional context, and minor revisions to strengthen the rigor and impact of the study.
1. Title and Abstract
Strength: The title is concise and informative. The abstract summarizes the study well.
Suggestion: The abstract would benefit from clearly listing the study design ('retrospective cohort') to align with best reporting practices.
Edit Suggestion: Include the exact sample size in the abstract for transparency.
2. Introduction
Strength: The background is well developed, and relevant literature is cited appropriately.
Concern: Some references are cited together repeatedly without differentiation.
Suggestion: Clarify rationale for focusing only on Black and Hispanic subgroups.
3. Methods
Strength: Clear description of data source, inclusion criteria, and statistical tests.
Major Weakness: No adjustment for confounders such as maternal age, BMI, etc.
Clarity: Structure description of analytical approach for better readability.
Minor Issue: Correct typographical errors (e.g., 'date' instead of 'data').
4. Results
Strength: Statistically robust and clearly tabulated.
Concern: Small differences (e.g., 0.8% vs. 0.3%) should be interpreted with caution.
Suggestion: Include confidence intervals along with p-values.
5. Discussion
Strength: Links results well to the 'healthy immigrant effect.'
Gap: Lack of country-of-origin and length-of-stay data should be emphasized as a limitation.
Future Directions: Propose concrete designs for future research.
6. Conclusions
Strength: The Conclusion is consistent with the results.
Suggestion: Explicitly note that causality cannot be inferred.
7. Tables and Figures
Strength: The Table is clear.
Improvement Needed: Add a figure to visualize subgroup differences.
8. References
Strength: Appropriate and recent references.
Technical Issue: Ensure formatting matches journal style.

Author Response
Reviewer 1
Open Review
(x) I would not like to sign my review report
( ) I would like to sign my review report
Quality of English Language
( ) The English could be improved to more clearly express the research.
(x) The English is fine and does not require any improvement.
|
Yes |
Can be improved |
Must be improved |
Not applicable |
|
|
Does the introduction provide sufficient background and include all relevant references? |
( ) |
(x) |
( ) |
( ) |
|
Is the research design appropriate? |
( ) |
(x) |
( ) |
( ) |
|
Are the methods adequately described? |
(x) |
( ) |
( ) |
( ) |
|
Are the results clearly presented? |
(x) |
( ) |
( ) |
( ) |
|
Are the conclusions supported by the results? |
(x) |
( ) |
( ) |
( ) |
|
Are all figures and tables clear and well-presented? |
( ) |
(x) |
( ) |
( ) |
Comments and Suggestions for Authors
Reviewer Comments to the Authors
General Assessment
This manuscript addresses an important topic in maternal health, specifically the prevalence of hypertensive disorders of pregnancy (HDP) with respect to nativity, race, and ethnicity. Using the PRAMS dataset allows for a robust and meaningful analysis that supports the concept of the 'healthy immigrant effect.' The manuscript is generally well written, and the results are clearly presented. However, some areas would benefit from clarification, additional context, and minor revisions to strengthen the rigor and impact of the study.
- Title and Abstract
Strength: The title is concise and informative. The abstract summarizes the study well.
Suggestion: The abstract would benefit from clearly listing the study design ('retrospective cohort') to align with best reporting practices.
Edit Suggestion: Include the exact sample size in the abstract for transparency.
A sample size of 63,648 was analyzed.
- Introduction
Strength: The background is well developed, and relevant literature is cited appropriately.
Concern: Some references are cited together repeatedly without differentiation.
Suggestion: Clarify rationale for focusing only on Black and Hispanic subgroups.
We added the following clarification in lines 41-70:
One of the aspects of maternal mortality in the United States and specifically preeclampsia that continues to be studied is its disproportionate impact on patients of racial and ethnic minorities3. The aforementioned morbidity and mortality report additionally showed a significantly higher proportion of pregnancy related deaths attributed to hypertensive disorders of pregnancy for black women than for white women1. These racial disparities exist to such an extent that “Black race” is listed as a moderate risk factor for developing preeclampsia on the ACOG (American College of Obstetricians and Gynecologists) website6,7. Alongside this racial disparity, there is a conversation about a difference in rates of preeclampsia between Hispanic women as compared to White women. Unlike the previous research regarding higher rates of preeclampsia among Black women, this research is a bit more variable8. While some studies show higher rates of preeclampsia among Hispanic women as compared to their White counterparts, others do not8. A study using data from New York City between the years of 1995-2003 broke down their Hispanic population into subgroups by region and found that Mexican women had the highest risk of preeclampsia among major ethnic groups in their study8,9. This is an example of a study which provides further insight into prevalence of preeclampsia among Hispanic women in the United States by examining Hispanic women by country rather than as a demographic monolith.
This study aims to conduct a deeper dive into rates of preeclampsia for women of racial and ethnic minorities who are living in the United States with a particular focus on women who were born outside of the United States but have moved to the US. Existing literature suggests the idea of the “healthy immigrant effect”---the phenomenon of new United States’ immigrants initially demonstrating better health metrics than their US-born counterparts, but declining as their tenure in the United States increases10. Previous literature indicates that the amount of time that immigrants who identify as Black and Hispanic spend in the United States has a greater negative impact on their health as compared to White and Asian immigrants10. This observed phenomenon leads to questions both about what aspects of health this effect can be observed in and about why this effect exists.
- Methods
Strength: Clear description of data source, inclusion criteria, and statistical tests.
Major Weakness: No adjustment for confounders such as maternal age, BMI, etc.
Clarity: Structure description of analytical approach for better readability.
Minor Issue: Correct typographical errors (e.g., 'date' instead of 'data').
Materials and Methods were rewritted and multiple additional clarifying statements were added to address the concern.
Multiple Manuscript reviewed and typographical errors identified and corrected, including ‘data’ instead of ‘date’ (autocorrect had changed this incorrectly so thanks for identifying).
- Results
Strength: Statistically robust and clearly tabulated.
Concern: Small differences (e.g., 0.8% vs. 0.3%) should be interpreted with caution.
Suggestion: Include confidence intervals along with p-values.
CI with p-values added accordingly along with OR:
- (OR 6.28 CI 5.84-6.76 for mothers not born in US having gestational HTN vs.mothers born in US p < .001). .
- OR 1.16 CI 0.92-1.46 for mothers not born in US vs. US born mothers having hypertension eclampsia; p = .197.
- OR 0.69 CI 0.59-0.82 for Black mothers not born in US to have GHTN vs. Black mothers born in US to have GHTN = 36.92, p < .001.
- OR 1.49 CI 0.88-2.53 for Black mothers not born in US having hypertension eclampsia vs. Black mothers born in US having hypertension eclampsia = 2.267, p = .132.
- OR 1.31 CI 1.14-1.51 for Hispanic mothers born in US vs. Hispanic mothers not born in US to have gestational hypertension = 18.236, p < .001.
- OR 2.34 CI 1.29-4.2 for Hispanic mothers born in US vs. Hispanic mothers not born in US to have diagnosis of hypertension eclampsia = 8.480, p = .004.
- Discussion
Strength: Links results well to the 'healthy immigrant effect.'
Gap: Lack of country-of-origin and length-of-stay data should be emphasized as a limitation.
Future Directions: Propose concrete designs for future research.
We added some statements to address concern as follows:
These questions remain unanswered by this study due to the limitations of the secondary data used for the analyses. This dataset did not include information about two variables of particular interest – length of tenure in the United States and country of origin.
Due to this limitation, Firstly, this study does not address how length of tenure in the United States impacts a person’s risk of developing hypertensive disorders of pregnancy. Previous studies about the healthy immigrant effect suggest a diminishing of this effect as time living in the US increases, with a greater decrease in the protective effect of foreign nativity for Black and Hispanic immigrants10. However, the referenced study was not specifically examining the health of pregnant patients. Moreover, it would be an interesting follow-up to the present study to examine hypertensive disorders of pregnancy in the context of length of tenure in the US for foreign-born mothers from racial and ethnic minority groups.
- Conclusions
Strength: The Conclusion is consistent with the results.
Suggestion: Explicitly note that causality cannot be inferred.
However, it is important to note that the relationships highlighted in this study cannot be used to infer causality due to the nature of the retrospective cohot methodology. In addition to contributing the direct results of this study, it It also provides further inspiration for future questions, some of which are outlined above.
- Tables and Figures
Strength: The Table is clear.
Improvement Needed: Add a figure to visualize subgroup differences.
4 additional figures were added to help visualize subgroup differences.
- References
Strength: Appropriate and recent references.
Technical Issue: Ensure formatting matches journal style.
Thanks so very much commentary. We have utilized journal style template and reviewed author guidelines and subsequently combed the manuscript to ensure this with multiple changes/corrections/edits.
Submission Date
16 May 2025
Date of this review
31 May 2025 14:00:56
Reviewer 2 Report
Comments and Suggestions for Authors
This paper examines associations of nativity and race/ethnicity with hypertensive disorders of pregnancy with data derived from PRAMS. The subject is relevant, especially considering maternal health disparities. The study is of high quality and the fact that it is a national data set gives further strength to the results. But there are some areas that really need to be edited for clarity and rewrite of interpretation.
- The inappropriate use of the word “hypertension eclampsia” is nonstandard, so may be replaced by “preeclampsia/eclampsia” or “hypertensive disorders of pregnancy (HDP)” in the text.
- It would be a good idea to provide an earlier, less nebulous definition of the “healthy immigrant effect” and of the hypothesized association with HDP in the Introduction.
- The Discussion could elaborate on non-significant results of the findings—such as among the Black subgroup—indicating whether (no) statistical difference was precluded due to power, strength of data, or similarity.
- Findings should be interpreted in the context of prior work (Boakye et al., Singh et al.). and interpret the results accordingly.
- In "Results," pair percentages with raw counts for clarity (e.g., 12.6% [n = 1,232]).
- Tables for gestational hypertension and preeclampsia could have been separated out, or more clearly labelled.
Here are some minor issues:
- “thereover” (l.236) should be emended to “moreover” or “furthermore”.
- Omit subjective language like “interestingly” (e.g., lns. 183, 216) Neutral academic style is better.
- Please be sure to replace all placeholder text (e.g., “date”) before publishing!
The current manuscript adds important information to the literature, particularly in addressing nativity-specific disparities in HDP.
Recommendation: Major Revision
Author Response
Reviewer 2
Open Review
(x) I would not like to sign my review report
( ) I would like to sign my review report
Quality of English Language
( ) The English could be improved to more clearly express the research.
(x) The English is fine and does not require any improvement.
|
Yes |
Can be improved |
Must be improved |
Not applicable |
|
|
Does the introduction provide sufficient background and include all relevant references? |
( ) |
(x) |
( ) |
( ) |
|
Is the research design appropriate? |
(x) |
( ) |
( ) |
( ) |
|
Are the methods adequately described? |
(x) |
( ) |
( ) |
( ) |
|
Are the results clearly presented? |
(x) |
( ) |
( ) |
( ) |
|
Are the conclusions supported by the results? |
( ) |
(x) |
( ) |
( ) |
|
Are all figures and tables clear and well-presented? |
( ) |
(x) |
( ) |
( ) |
Comments and Suggestions for Authors
This paper examines associations of nativity and race/ethnicity with hypertensive disorders of pregnancy with data derived from PRAMS. The subject is relevant, especially considering maternal health disparities. The study is of high quality and the fact that it is a national data set gives further strength to the results. But there are some areas that really need to be edited for clarity and rewrite of interpretation.
- The inappropriate use of the word “hypertension eclampsia” is nonstandard, so may be replaced by “preeclampsia/eclampsia” or “hypertensive disorders of pregnancy (HDP)” in the text.
Response:
We included a statement regarding terminology in the methods to explain non-standard usage: “this paper utilizes the nonstandard term ”hypertension eclampsia” because this is the term utilized in the PRAMS dataset from which this study’s conclusions were drawn.”
healthy immigrant effect”---the phenomenon of new United States’ immigrants initially demonstrating better health metrics than their US-born counterparts, but declining as their tenure in the United States increases10
Hamilton TG, Hagos R. Race and the Healthy Immigrant Effect. Public Policy Aging Rep. 2021;31(1):14-18. doi:10.1093/ppar/praa042.
- It would be a good idea to provide an earlier, less nebulous definition of the “healthy immigrant effect” and of the hypothesized association with HDP in the Introduction.
Response
We agree in concept that the term “healthy immigrant” may trigger sensitivity, noting that this is a term that is recently being utilized and has been defined by respected authorship that has notably been generated with intent to be sensitive, thoughtful and respectful. Highlighting these effects are made with intent to identify undesired impacts from public policies that restrict access and have negative impacts on health of immigrant communities to social services designed to improve public health. We have included the following statement to this effect:
- Findings should be interpreted in the context of prior work (Boakye et al., Singh et al.). and interpret the results accordingly.
Response:
Looking specifically at the first subsample examined in this study – participants who self-identified their race as Black – the finding appears to replicate findings in the previous study referenced earlier12. Like this study, the Boakye et al. study found lower prevalence of a hypertensive disorder of pregnancy (their study specifically looked at preeclampsia) in their foreign-born sample of Black women as compared to their US-born counterparts12. A piece of information that the present study adds to the conversation is this replicated effect using a national dataset as Boakye et al. utilize a dataset from the Boston Birth Cohort. The present study’s finding of a lower prevalence of a hypertensive disorder of pregnancy amongst Black foreign-born women as compared to their US-born counterparts on a national level is reflective of the work of Singh et al. in their study using national birth cohort data between 2014 and 201511. Overall, this study adds further evidence suggesting that the healthy immigrant effect can be appropriately applied to explain higher rates of hypertensive disorders of pregnancy for US-born Black women than for their foreign-born counterparts.
- The Discussion could elaborate on non-significant results of the findings—such as among the Black subgroup—indicating whether (no) statistical difference was precluded due to power, strength of data, or similarity.
Response:
The analysis with this group was the only one which yielded a significant difference in prevalence of hypertension eclampsia between the foreign-born participants and that of their US-born counterparts. The overall subset of participants with hypertension eclampsia (460) is much smaller than the subset of participants with gestational hypertension (6,535). Because of this, it is difficult to make a strong conclusion based off of the significantly greater proportion of US-born Hispanic women with hypertension eclampsia as compared to their foreign-born counterparts in the context of the lack of a significant difference for this comparison in our groups of Black women and of the full dataset of women. It does suggest a potentially uniquely lower risk of hypertension eclampsia for foreign-born Hispanic mothers, but further research should be done before this conclusion is made.
- In "Results," pair percentages with raw counts for clarity (e.g., 12.6% [n = 1,232]).
We included n’s with paired percentages for all data points in the tables.
- Tables for gestational hypertension and preeclampsia could have been separated out, or more clearly labelled.
Table 2 added with attempt to clearly label.
|
|
HTN Eclampsia |
No HTN Eclampsia |
|
Full Dataset: |
|
|
|
US-born |
369 (0.7%) |
50985 (99.3%) |
|
Foreign-born |
91 (0.8%) |
10806 (99.2%) |
|
Black Race: |
|
|
|
US-born |
59 (0.6%) |
9568 (99.4/%) |
|
Foreign-born |
18 (0.9%) |
1948 (99.1%) |
|
Hispanic Ethnicity: |
|
|
|
US-born |
52 (0.8%)* |
6794 (99.2%) |
|
Foreign born |
14 (0.3%)* |
4292 (99.7%) |
Here are some minor issues:
- “thereover” (l.236) should be emended to “moreover” or “furthermore”.
Moreover, it would be an interesting follow-up to the present study to examine hypertensive disorders of pregnancy in the context of length of tenure in the US for foreign-born mothers from racial and ethnic minority groups.
- Omit subjective language like “interestingly” (e.g., lns. 183, 216) Neutral academic style is better.
All instances (n=2) of ‘interestingly’ were removed (lines 214 and 248).
- Please be sure to replace all placeholder text (e.g., “date”) before publishing!
The submission date (May 16, 2025) and revision dates (June 22, 2025) were added accordingly.
The current manuscript adds important information to the literature, particularly in addressing nativity-specific disparities in HDP.
Response: Thank you so very much for constructive review and feedback. We have attempted to address all concerns. Thank you.
Recommendation: Major Revision
Submission Date
16 May 2025
Date of this review
05 Jun 2025 11:33:50
Reviewer 3 Report
Comments and Suggestions for Authors
Dear authors,
Please tabulate the information in lines 123-130.
Please edit table 1 to include the n of US born women and foreign born women included in the study.
Correct the symbol for chi-squared test throughout the paper.
Is there any information in literature comparing the prevalence of preeclampsia in white women (US born vs foreign born)? It would be useful to add that to the discussion.
Author Response
Reviewer 3
Open Review
(x) I would not like to sign my review report
( ) I would like to sign my review report
Quality of English Language
( ) The English could be improved to more clearly express the research.
(x) The English is fine and does not require any improvement.
|
Yes |
Can be improved |
Must be improved |
Not applicable |
|
|
Does the introduction provide sufficient background and include all relevant references? |
(x) |
( ) |
( ) |
( ) |
|
Is the research design appropriate? |
( ) |
(x) |
( ) |
( ) |
|
Are the methods adequately described? |
(x) |
( ) |
( ) |
( ) |
|
Are the results clearly presented? |
( ) |
(x) |
( ) |
( ) |
|
Are the conclusions supported by the results? |
(x) |
( ) |
( ) |
( ) |
|
Are all figures and tables clear and well-presented? |
( ) |
( ) |
(x) |
( ) |
Comments and Suggestions for Authors
Dear authors,
Please tabulate the information in lines 123-130.
We added the following tabulation:
Table 2. HYPERTENSION ECLAMPSIA PREVALENCE AMONGST FOREIGN AND US-BORN MOTHERS.
|
|
HTN Eclampsia |
No HTN Eclampsia |
|
Full Dataset: |
|
|
|
US-born |
369 (0.7%) |
50985 (99.3%) |
|
Foreign-born |
91 (0.8%) |
10806 (99.2%) |
|
Black Race: |
|
|
|
US-born |
59 (0.6%) |
9568 (99.4/%) |
|
Foreign-born |
18 (0.9%) |
1948 (99.1%) |
|
Hispanic Ethnicity: |
|
|
|
US-born |
52 (0.8%)* |
6794 (99.2%) |
|
Foreign born |
14 (0.3%)* |
4292 (99.7%) |
Please edit table 1 to include the n of US born women and foreign born women included in the study.
Edited to include N and %:
|
US-born |
5709 (11.0%)* |
46284 (89.0%) |
|
Foreign-born |
816 (7.0%)* |
10839 (93.0%) |
|
Black Race: |
|
|
|
US-born |
1232 (12.6%)* |
8549 (87.4%) |
|
Foreign-born |
182 (8.0%)* |
2083 (92.0%) |
|
Hispanic Ethnicity: |
|
|
|
US-born |
663 (9.5%)* |
6324 (90.5%) |
|
Foreign born |
327 (7.2%)* |
4210 (92.8%) |
Correct the symbol for chi-squared test throughout the paper.
Χ2 added throughout; n=11.
Is there any information in literature comparing the prevalence of preeclampsia in white women (US born vs foreign born)? It would be useful to add that to the discussion.
We included/added the following to our introduction as key background to the justification for the study:
- A study using data from New York City between the years of 1995-2003 broke down their Hispanic population into subgroups by region and found that Mexican women had the highest risk of preeclampsia among major ethnic groups in their study8,9. This is an example of a study which provides further insight into prevalence of preeclampsia among Hispanic women in the United States by examining Hispanic women by country rather than as a demographic monolith.
- A study using the 2014-2015 national birth cohort data found evidence of lower rates of maternal hypertension amongst immigrant women as compared to US-born women11. A 2021 study conducted by Boakye et al. examines the prevalence of preeclampsia in foreign-born non-Hispanic Black women as compared to US-born non-Hispanic Black women, and found lower rates in the population of foreign-born non-Hispanic black women12
We additionally included these in the discussion to place our findings/results/implications in context of prior studies:
- Like this study, the Boakye et al. study found lower prevalence of a hypertensive disorder of pregnancy (their study specifically looked at preeclampsia) in their foreign-born sample of Black women as compared to their US-born counterparts12. A piece of information that the present study adds to the conversation is this replicated effect using a national dataset as Boakye et al. utilize a dataset from the Boston Birth Cohort. The present study’s finding of a lower prevalence of a hypertensive disorder of pregnancy amongst Black foreign-born women as compared to their US-born counterparts on a national level is reflective of the work of Singh et al. in their study using national birth cohort data between 2014 and 201511. Overall, this study adds further evidence suggesting that the healthy immigrant effect can be appropriately applied to explain higher rates of hypertensive disorders of pregnancy for US-born Black women than for their foreign-born counterparts.
- The second subsample examined in this study is made up of participants who self-identified their ethnicity as Hispanic. As mentioned before, there is less consensus about the relationship between Hispanic ethnicity and prevalence of hypertensive disorders of pregnancy8. Some previous work has shown evidence of higher risks of preeclampsia for subgroups of Hispanic women8,9. However, other prior work has concluded that disparities in prevalence of preeclampsia could not be attributed to nativity for Hispanic women13. The present study adds to this discussion by providing evidence of significantly lower rates of both gestational hypertension and hypertension eclampsia for foreign-born Hispanic women than for their US-born counterparts.
Thank you very much for the constructive review and commentary. We have attempted to address all concerns.
Submission Date
16 May 2025
Date of this review
08 Jun 2025 03:11:37
Round 2
Reviewer 2 Report
Comments and Suggestions for Authors
Thank you for addressing the previous concerns with clarity and detail. The revised manuscript shows clear improvement in terminology, structure, and interpretive depth.
Strengths of the Revision:
-
The authors appropriately clarified the use of the term “hypertension eclampsia” as reflecting PRAMS terminology, which resolves prior confusion.
-
Effect sizes and confidence intervals (OR, CI) were added to supplement Chi-square tests, which greatly enhance interpretability.
-
Tables were split to better distinguish outcomes for gestational hypertension and eclampsia—this is a helpful change.
-
The Discussion section was expanded to better contextualize findings within previous literature (e.g., Boakye et al., Singh et al.) and address study limitations.
Remaining Suggestions for Improvement:
-
Causal language: In the conclusion of the Discussion, please ensure the language avoids suggesting causality (e.g., “effect of nativity” should be “association with nativity” due to the retrospective study design).
-
Minor clarity issues:
-
The phrase “hypertension eclampsia” still appears awkward despite the PRAMS explanation. Consider introducing a brief phrase such as: “herein referred to as ‘hypertension eclampsia’ based on PRAMS item labels.”
-
Check sentence in line 287–291: “...it also provides further inspiration...” might be better stated as “...it also raises important future research questions...”
-
-
Typographical error: In Table 2 (line 200), the label “No HTN Eclamp-sia” contains a hyphenation error.
These are minor, non-substantive issues. Overall, the manuscript is much improved and now clearly communicates its methods and findings.
Author Response
Open Review
(x) I would not like to sign my review report
( ) I would like to sign my review report
Quality of English Language
( ) The English could be improved to more clearly express the research.
(x) The English is fine and does not require any improvement.
|
Yes |
Can be improved |
Must be improved |
Not applicable |
|
|
Does the introduction provide sufficient background and include all relevant references? |
(x) |
( ) |
( ) |
( ) |
|
Is the research design appropriate? |
(x) |
( ) |
( ) |
( ) |
|
Are the methods adequately described? |
(x) |
( ) |
( ) |
( ) |
|
Are the results clearly presented? |
(x) |
( ) |
( ) |
( ) |
|
Are the conclusions supported by the results? |
(x) |
( ) |
( ) |
( ) |
|
Are all figures and tables clear and well-presented? |
(x) |
( ) |
( ) |
( ) |
Comments and Suggestions for Authors
Thank you for addressing the previous concerns with clarity and detail. The revised manuscript shows clear improvement in terminology, structure, and interpretive depth.
Strengths of the Revision:
- The authors appropriately clarified the use of the term “hypertension eclampsia” as reflecting PRAMS terminology, which resolves prior confusion.
- Effect sizes and confidence intervals (OR, CI) were added to supplement Chi-square tests, which greatly enhance interpretability.
- Tables were split to better distinguish outcomes for gestational hypertension and eclampsia—this is a helpful change.
- The Discussion section was expanded to better contextualize findings within previous literature (e.g., Boakye et al., Singh et al.) and address study limitations.
Remaining Suggestions for Improvement:
- Causal language: In the conclusion of the Discussion, please ensure the language avoids suggesting causality (e.g., “effect of nativity” should be “association with nativity” due to the retrospective study design).
Response:
Language reviewed and edited to avoid causality in discussion with emphasis on this observation being an association (eg, 1. lines 271-275: Previous studies about the healthy immigrant effect suggest diminished association with better health in foreign born persons as time living in the US increases, with a greater decrease in the apparent protection or association observed with foreign nativity for Black and Hispanic immigrants10. 2. lines 65-68: This observed phenomenon leads to questions both about what aspects of health this effect can be observed in and about why this association exists. 3. Lines 295-297: Overall, the present study echoes evidence of the healthy immigrant effect in pregnant mothers as seen in prior research and adds specific insight into this association as it relates to hypertensive disorders of pregnancy.
- Minor clarity issues:
- The phrase “hypertension eclampsia” still appears awkward despite the PRAMS explanation. Consider introducing a brief phrase such as: “herein referred to as ‘hypertension eclampsia’ based on PRAMS item labels.”
Response; lines 112-117 were edited in methods as follows:
Subjects were stratified by response ‘yes’ or ‘no’ to having gestational hypertension and hypertension eclampsia to generate the proportions. Note that this paper utilizes the nonstandard term ”hypertension eclampsia” because this is the term utilized in the PRAMS dataset from which this study’s conclusions were drawn. Accordingly, eclampsia is herein referred to as hypertension eclampsia based on PRAMS item labels.
- Check sentence in line 287–291: “...it also provides further inspiration...” might be better stated as “...it also raises important future research questions...”
Lines 296-301 were revised as follows:
Overall, the present study echoes evidence of the healthy immigrant effect in pregnant mothers as seen in prior research and adds specific insight into this association as it relates to hypertensive disorders of pregnancy. However, it is important to note that the relationships highlighted in this study cannot be used to infer causality due to the nature of the retrospective cohort methodology. In addition to contributing the direct results of this study, it Lastly, our study raises important questions that warrant consideration in future research efforts.
- Typographical error: In Table 2 (line 200), the label “No HTN Eclamp-sia” contains a hyphenation error.
Response: Table resized to avoid this issue. We agree that was awkward and hopefully looks better now as follows:
|
|
HTN Eclampsia |
No HTN Eclampsia |
|
Full Dataset: |
|
|
|
US-born |
369 (0.7%) |
50985 (99.3%) |
|
Foreign-born |
91 (0.8%) |
10806 (99.2%) |
|
Black Race: |
|
|
|
US-born |
59 (0.6%) |
9568 (99.4/%) |
|
Foreign-born |
18 (0.9%) |
1948 (99.1%) |
|
Hispanic Ethnicity: |
|
|
|
US-born |
52 (0.8%)* |
6794 (99.2%) |
|
Foreign born |
14 (0.3%)* |
4292 (99.7%) |
These are minor, non-substantive issues. Overall, the manuscript is much improved and now clearly communicates its methods and findings.
Response: Thanks you very much for providing highly constructive and useful review. We have attempted to address all concerns and revised accordingly in the attached files. Thanks again.
Submission Date
16 May 2025
Date of this review
24 Jun 2025 23:19:49